# Geographic Object-Based Image Analysis for Automated Landslide Detection Using Open Source GIS Software

**Raphael Knevels [1],\***  **, Helene Petschko [1]** , **Philip Leopold [2] and Alexander Brenning [1]**

1   Friedrich Schiller University Jena, Department of Geography, D JENA01 Jena, Germany;
    helene.petschko@uni-jena.de (H.P.); alexander.brenning@uni-jena.de (A.B.)
2   AIT Austrian Institute of Technology GmbH, Center for Mobility Systems, 1220 Vienna, Austria;
    philip.leopold@ait.ac.at
\*   Correspondence: raphael.knevels@uni-jena.de

**Abstract:** With the increased availability of high-resolution digital terrain models (HRDTM) generated using airborne light detection and ranging (LiDAR), new opportunities for improved mapping of geohazards such as landslides arise. While the visual interpretation of LiDAR, HRDTM hillshades is a widely used approach, the automatic detection of landslides is promising to significantly speed up the compilation of inventories. Previous studies on automatic landslide detection often used a combination of optical imagery and geomorphometric data, and were implemented in commercial software. The objective of this study was to investigate the potential of open source software for automated landslide detection solely based on HRDTM-derived data in a study area in Burgenland, Austria. We implemented a geographic object-based image analysis (GEOBIA) consisting of (1) the calculation of land-surface variables, textural features and shape metrics, (2) the automated optimization of segmentation scale parameters, (3) region-growing segmentation of the landscape, (4) the supervised classification of landslide parts (scarp and body) using support vector machines (SVM), and (5) an assessment of the overall classification performance using a landslide inventory. We used the free and open source data-analysis environment R and its coupled geographic information system (GIS) software for the analysis; our code is included in the Supplementary Materials. The developed approach achieved a good performance ($\kappa$ = 0.42) in the identification of landslides.

**Keywords:** geographic object-based image analysis; GEOBIA; open source GIS; landslide detection; LiDAR; high-resolution digital terrain model; HRDTM; support vector machine; SVM

## 1. Introduction

Landslides are natural phenomena occurring worldwide. With the ongoing environmental change and the potential exposure of humans to landslides, the identification of areas susceptible to landsliding has become increasingly important for risk reduction and spatial planning [1,2].

Landslide inventories are a fundamental source of information for the creation of landslide susceptibility maps [3]. The quality of inventories is of critical importance for the explanatory power and unbiasedness of the landslide susceptibility models and their predictions [4], but their compilation is time-consuming and hinges on the subjective assessment of experts [3,5]. During the last decade, the traditional creation of landslide inventories by visual interpretation of aerial photographs and extensive fieldwork has been increasingly supported by or even replaced with the expert-based visual detection or (semi- or fully) automated landslide detection using light detection and ranging (LiDAR) data [3,6,7]. In particular, the increasing availability of airborne LiDAR-derived high-resolution digital terrain models (HRDTM) creates the opportunity to detect landslides even within forests, where

imagery from passive optical sensors is of limited utility [7], or in remote, difficult-to-access areas such as high mountains [6].

When visually mapping landslides, hillshade maps, contour lines, or slope maps are used for interpreting landslide morphology [3,8]. Focusing only on HRDTM-derived data for semi- or fully automated landslide detection, several promising techniques have emerged in recent years: Booth et al. [9] and Kasai et al. [10] used filter techniques such as Fourier and wavelet transform [9] or eigenvalue ratios [10] to detect landslides based on their topographic signature. Leshchinsky et al. [11] and Bunn et al. [12] developed and extended the contour connection method, a vector-based method to detect different landslide features using a mesh of nodes and connecting lines. Pawłuszek et al. [13] detected landslides using a pixel-based classification scheme, and benchmarked different classifiers while also accounting for different grid resolutions. However, with high-resolution data, pixel-based classification approaches are often confronted with noise inherent in the data ("salt-and-pepper effect"), resulting in a high false positive rate. Geographic object-based image analysis (GEOBIA) tries to overcome this challenge by segmenting the image initially into regions with similar characteristics, mimicking the way humans perceive their environment [14]. GEOBIA therefore promises to delineate objects that match an entire landslide polygon or specified landslide parts (scarp, body, flank, deposit, etc.).

GEOBIA has already demonstrated its potential in landslide mapping [7,15]. Commonly, high-resolution optical and multispectral data are combined with HRDTM derivatives for landslide detection [6,16–19]. In these studies, the segmentation is performed on optical data or on derived indices such as the NDVI (normalized difference vegetation index). The aim of these analyses was the detection of each landslide by one single object under consideration of the landslides post-failure characteristics [7,20] (see Section 2.3). However, these approaches have the drawback of failing to detect landslides in forests, and of relying on recent observations of landslide events and on cloud-free optical data. Until now, only few authors based landslide detection solely on LiDAR HRDTM-derived variables [7,15,21]. The use of LiDAR HRDTM-derived data avoids the above-mentioned limitations of optical remote-sensing imagery (cloud-free data, landslides under forest). However, the geomorphometric structure of a landslide is too complex to be represented by one single object. Instead, each landslide part must be identified using its specific geomorphometric characteristics on an appropriate scale [7].

Moreover, previous landslide detection procedures were mostly implemented in commercial software environments such as Definiens eCognition [7,15,16,18,21], ESRI ArcGIS [13,15], the ENVI image analysis software [13,15], or MathWorks MATLAB [9,21]. Only few authors made their algorithms publicly available [9,18], or used computing environments released under open source licenses [11]. To our knowledge no previous publication has presented open-access procedures for landslide detection that are entirely implemented in open source software environments and only require HRDTM-derived data. This is surprising, as the development and publication of algorithms in an open source environment enhances research accessibility, reproducibility, and transparency.

In this study, we examined the potential of open source software to conduct GEOBIA for landslide detection using solely HRDTM-derived variables. We combined land-surface variables, textural features, shape metrics and adjacencies to characterize and segment landslide parts, and subsequently identified landslide scarps and bodies using a support vector machine (SVM). Our study area was located in a landslide-prone area in Burgenland, Austria, where a HRDTM with a $1 \times 1$ m resolution was available. We mapped a historical landslide inventory (delineating the scarp and body) by visual interpretation of the LiDAR HRDTM. Since in GEOBIA, the segmentation and the classification techniques are subject to different optimization procedures, we used a rigorous statistical approach to select optimal segmentation scale parameters and classifier hyperparameters. The workflow was implemented in the free and open source programming environment R using the open source GIS (geographic information system) software for geodata processing. Sample data and code can be found in the Supplementary Materials.

## 2. Materials and Methods

### 2.1. Study Area and Data

The study area in Austria's easternmost province Burgenland is located in the district of Oberpullendorf (municipalities Piringsdorf, Pilgersdorf, Unterragbnitz-Schwendgraben, and Steinberg-Dörfl). It covers 111 km² and extends from 16°17′ E 47°31′ N to 16°31′ E 47°24′ N. We split the study area based on administrative boundaries into two similarly large regions with comparable geographic characteristics: Piringsdorf and Steinberg-Dörfl were used for training, while Pilgersdorf and Unterrabnitz-Schwendgraben were used as the validation area (Figure 1).

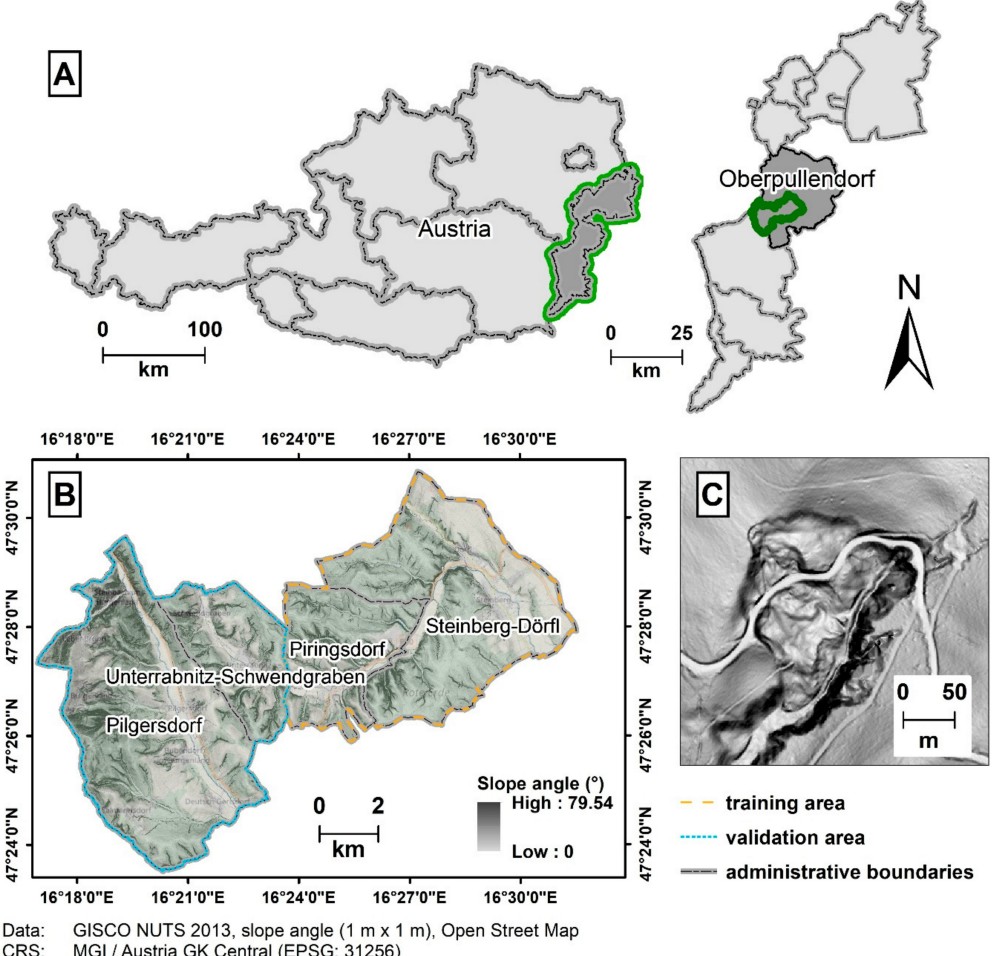

**Figure 1.** Overview of the study area: (**A**) Location in the district of Oberpullendorf and Austria; (**B**) study area split into training and validation areas; (**C**) example of a landslide visible in a slope map.

Over the last decade, the government of Burgenland recorded an increasing frequency of landslide-related damages to buildings and infrastructure. This has been attributed to an increased frequency of heavy precipitation events, as well as construction activities in landslide-prone areas [22]. The predisposition to landsliding is furthermore related to the underlying geology, which is dominated by basin sediments of Neogene and Quaternary age mainly consisting of clay, silt, sand, and gravel.

Our analysis was based on an airborne LiDAR-derived HRDTM with a 1 × 1 m resolution, which was provided by the Government of Burgenland (survey dates 23 November 2010 and 10 April 2011). It was created using the RIEGL LMS-Q560 and LMS-Q650 laser scanners (providing on average more than 10 last-return points per m²) and Terrasolid's software TerraModeler.

We created a historical landslide inventory based on the available LiDAR HRDTM using visual mapping methods described in [8]. Using the classification scheme introduced by Cruden and Varnes [20], we focused on mapping landslides in earth and debris materials of the slide type with possible transitions to complex slide flows. Furthermore, we distinguished between landslide main scarp and landslide body where discernible. Minor scarps or cracks located downslope of the main scarp and inside the landslide body, were not recorded. Overall, 382 landslides were digitized in vector format in the study area (training area: 198 landslides, validation area: 184 landslides; refer to Table A1 in the Appendix A for more information on the inventory).

## 2.2. Open Source Software and R

Free and open source software has been gaining popularity in all scientific domains in the last decade as one important aspect of open science [23]. With the establishment of the Open Source Geospatial Foundation (OSGeo) in 2006, the movement of open source reached the GIS community, and free software and algorithms are increasingly published under open source licenses [23]. Our study was fully conducted in the free and open source development environment R [24]. R's major strengths include the availability of numerous advanced statistical analysis methods, as well as the possibility to easily share and extend R functionalities through user-contributed packages [25]. Many packages for handling and analyzing geospatial data and integrating external GIS software are available and continuously under development [25,26].

For the steps in our workflow that required GIS capabilities, we considered GIS software that could be easily coupled with R and had suitable terrain analysis capabilities. We decided to use GRASS (Geographic Resources Analysis Support System) GIS 7.2.0 [27] and SAGA (System for Automated Geoscientific Analysis) GIS 6.3.0 [28] for geomorphometric operations and image segmentation. The geocomputing tools available in these open source desktop GIS can be accessed from within R using the packages *rgrass7* [29], and *RSAGA* [30], respectively. In addition, we used TauDEM 5.3 (Terrain Analysis Using Digital Elevation Models) [31] and the RVT 1.3 (Relief Visualization Toolbox) [32]. Since R packages that interface RVT 1.3 and TauDEM 5.3 are not yet available, we executed their tools through operating system commands in R.

As classification environment we used the *mlr* package [33]. *mlr* provides an easy-to-use framework for the most commonly used statistical and machine-learning models in R. Furthermore, it supports hyperparameter tuning, feature selection, variable importance assessment, and model performance evaluation even in parallelized mode.

The study was performed with the R version 3.5.3. The code is included in the Supplementary Materials.

## 2.3. GEOBIA of Landslides

GEOBIA reduces the level of detail and complexity inherent in an image by segmenting, clustering, or regionalizing it into distinguishable "meaningful objects" [34,35]. These objects represent real-world objects, emulating human perception [36]. Furthermore, objects carry various attributes such as area, shape, textural features, or spatial relationships between objects, and other statistical information [34].

In manual landslide mapping, experts often base their interpretation on typical morphological characteristics discernable in hillshade maps, contour lines, or slope maps [3,8]. Although landslides are complex geomorphological structures, their scarps and bodies are often clearly distinguishable in a slope or hillshade map. A landslide scarp is typically characterized by its steep slope and semi-circular form, adjacent to an often hummocky topography or rough surface downslope—the landslide body. Eeckhaut et al. [7] provided a detailed conceptualization of a landslide and all its parts in the context of GEOBIA.

In our analysis we focused on detecting landslides based on a slope angle segmentation and a classification including LiDAR-derived land-surface variables, textural features, shape metrics, and adjacencies. Due to the complexity of landslides, it is not possible to detect a landslide as one single,

homogeneous object. Instead, in our study a landslide was recognized based on its scarp and body in a three-step 'mask-segment-classify' procedure (Figure 2): (1) In the 'mask' step, we generated two disjoint masks containing candidate scarp and body areas, respectively; we chose high-pass filtering and thresholding techniques for this step (see Section 2.3.1). (2) In the 'segment' step, we segmented both masks separately into small, homogeneous objects, and merged these objects into a final segmentation output (see Section 2.3.2). (3) In the final 'classify' step, we extracted features describing the objects (see Section 2.3.3), detected landslides using a SVM classifier and assessed the model's variable importance based on permutation (see Section 2.3.4). We applied a post-processing procedure on the classification (see Section 2.3.5), and evaluated the results using pixel- and object-level accuracy assessments (see Section 2.3.6).

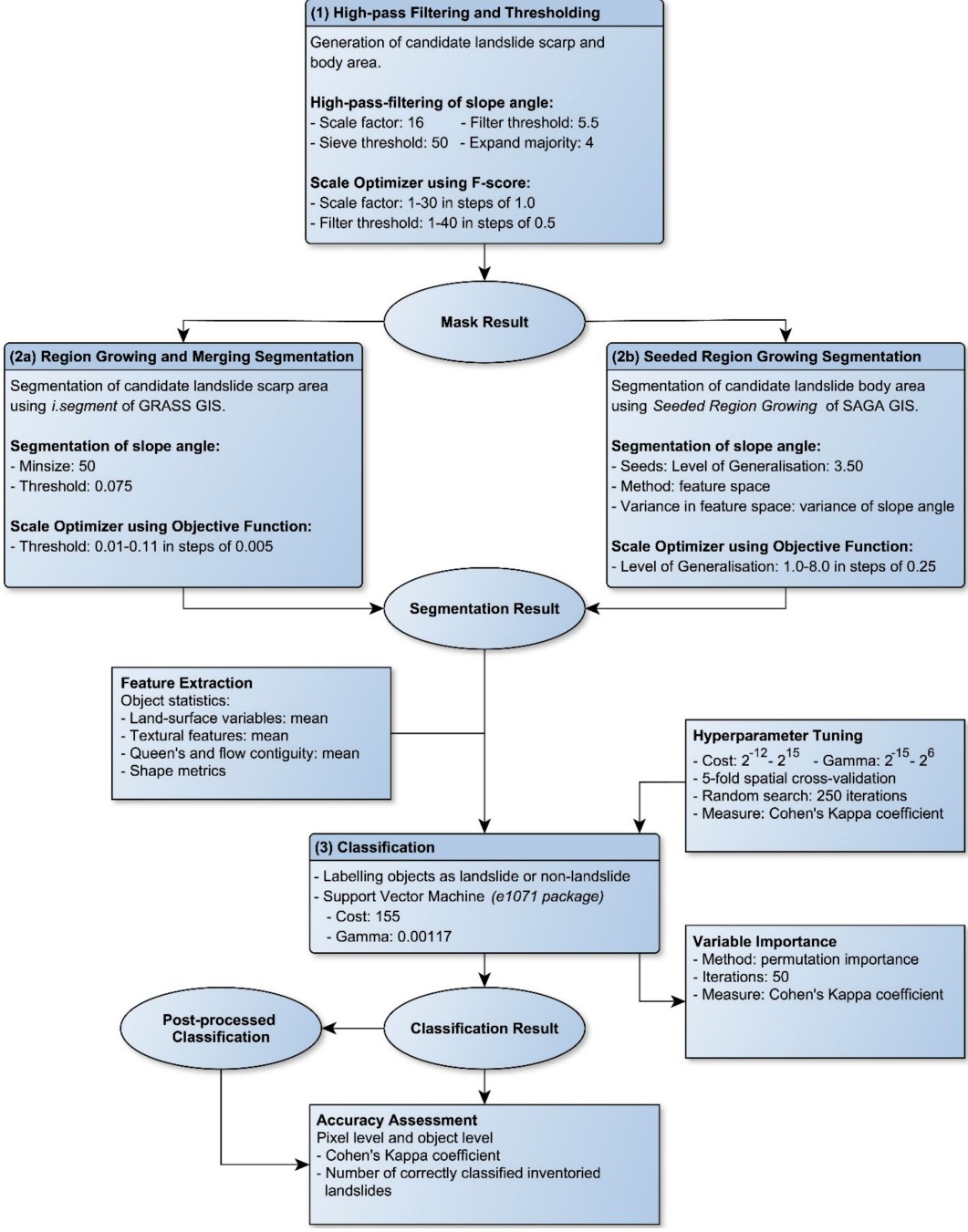

**Figure 2.** Overview of the proposed geographic object-based image analysis approach for landslide detection.

### 2.3.1. Masking: Generation of Candidate Landslide Scarp and Body Area

To account for the different underlying morphological complexity, we discriminated the study area into candidate landslide scarp and body areas. While a scarp can be recognized by a steep slope, there is still no strong and unique land-surface variable describing the complex structure of a landslide body [37]. Based on the assumption that the landslide scarp clearly stands out from its surroundings, we high-pass filtered the local slope angle and thresholded the result to select possible scarp areas. The remaining area was further considered for the identification of possible landslide bodies. This step was performed in SAGA GIS.

Two hyperparameters control the filtering and thresholding: A scale factor $S$ controlling the degree of smoothing in the high-pass filtering, and a threshold $T$ determining the minimum slope angle above which a grid cell was further considered as candidate scarp. We optimized $S$ and $T$ using a grid search (step (1) in Figure 2). We used the F-score to compare the suitability of $S$ and $T$ parameters. The F-score with $\beta = 1$ is defined as harmonic average of precision and recall (also known as F1-score) [38]. By increasing $\beta$, recall gets $\beta$ times as much importance as the precision. We set $\beta$ to 3 in this first screening step in order to avoid discarding candidate landslide scarps. Optimal $S$ and $T$ values were chosen using only the training data; the tuned parameter values were later applied to the validation area.

### 2.3.2. Segmentation

In GEOBIA the segmentation is a crucial processing step. Each segmentation algorithm requires so-called scale parameters influencing the shape and size of the resulting objects. The scale parameters vary depending on the applied segmentation algorithm. Moreover, the selection of appropriate scale parameters is a challenging task and is often estimated subjectively using trial-and-error methods [39].

There are several approaches for objectively estimating scale parameters by the use of an objective function (OF) [39,40]. In our study, we estimated optimal scale parameters based on the OF developed by Espindola et al. [40] and its extension by Martha et al. [18]. The OF is based on the idea that the resulting segmentation objects should have two desirable properties: First, internal homogeneity as measured by a variance indicator, and second, external heterogeneity as measured through the spatial autocorrelation index Moran's I. The plateau objective function (POF) developed by Martha et al. [18] aims to find suitable scale parameters from a set of nearly optimal candidate values in order to better represent the specific characteristics of the targeted object type. In this approach, local optima of the OF that are within a specific range of the OF's optimum, are considered as candidate values of the scale parameter [18]. We selected a suitable scale parameter from these candidate values based on visual inspection of the resulting object geometry. As in the previous step, this optimization and parameter selection took place on the training set, and the tuned parameters were later applied to the validation area.

For the segmentation of the candidate scarp and body areas, we examined the algorithms available in SAGA and GRASS GIS. The segmentation in SAGA GIS runs faster than its GRASS GIS counterpart due to an internal parallelization; it is therefore especially suitable for large areas at high resolutions. However, the SAGA GIS segmentation was not able to ignore no-data areas (i.e., the areas masked out in the first step), which created artefacts in the output and increased execution time. As a workaround, we decided to combine the segmentation results of GRASS and SAGA GIS in one segmentation output: The segmentation of the candidate scarp area (5% of study area, widespread no-data, computationally less demanding) was performed using the region growing algorithm of GRASS GIS (see [41] for more details). As a scale parameter the *threshold*—determining the allowed homogeneity distinction for merging adjacent pixel to an object—was optimized with the OF (step (2a) in Figure 2). For the segmentation of the larger candidate landslide body area (95% of study area, less no-data, computationally intensive), we applied the seeded region growing algorithm of SAGA GIS (see [42] for more details). Here, starting points, so-called seeds, must be defined, from which the growth of the objects starts. Their number and locations influence the applicability and quality of the resulting objects. SAGA GIS determines suitable seeds by a geostatistical approach [43]. Therefore,

only the controlling scale parameter, the *Level of Generalisation*, was optimized using the OF (step (2b) in Figure 2).

Segmentation was performed on the slope angle raster. We arbitrarily set a minimum object size of 50 m$^2$ to avoid over-segmentation. In addition, we accepted that an inventoried landslide body or scarp can be represented by single or multiple objects. Further settings are shown in Figure 2.

### 2.3.3. Feature Extraction for Classification

Based on the HRDTM various descriptive measures—so-called land-surface parameters or variables, can be extracted to describe the Earth surface [54]. We reviewed literature for relevant land-surface variables considering also the need to compute them in open source software. The selected, mostly primary land-surface variables are slope angle, curvature (maximum, minimum, plan, profile), normalized height, openness, sky-view factor, flow direction, flow accumulation, and roughness index. Furthermore, we calculated two topographically-guided textural features (entropy and standard deviation) by the use of the flow direction following the recommendations of Stumpf and Kerle [19]. However, because of limitations of the software used, unlike [19] we calculated the textural features at the pixel level, in the first step, followed by taking the average for each object, in the second step. GRASS GIS's *r.texture* implementation was used with the settings as specified in Table 1.

**Table 1.** Features used in this study.

| Variable | Software | Setting | Method |
|---|---|---|---|
| *land-surface variables* | | | |
| curvature (maximum, minimum, plan, profile) | SAGA GIS | WS 15 | [44] |
| flow direction, D-Infinity * | TauDEM | | [45] |
| flow accumulation, D-Infinity | TauDEM | log-transformation | [45] |
| normalized height | SAGA GIS | w = 5; t = 2; e = 2 | [46] |
| Openness | SAGA GIS | WS 50 | [7] |
| roughness index | SAGA GIS | WS 15 | [47] |
| sky-view factor | RVT | SR 20; D 16 | [32] |
| slope angle | SAGA GIS | | [48] |
| *textural features in flow direction* | | | |
| Entropy | GRASS GIS | WS 5; Dst 1 | [19,49] |
| standard deviation | GRASS GIS | WS 5; Dst 1 | [19,49] |
| *shape metrics* | | | |
| shape index | SAGA GIS | | [50] |
| interior edge ratio | SAGA GIS | | [50] |
| maximum distance between a polygon's vertices | SAGA GIS | | [50] |
| Compactness | R | | [51] |
| Convexity | R | | [52] |
| length-to-width ratio | R | | [53] |
| object orientation to flow direction | R | | [7,53] |

\*: Not used in classification but to determine the 'object orientation to flow direction'; Setting, scale-dependent parameters: WS: Size of processing window; Dst: Distance in textural feature calculation; SR, D: Search radius (SR) and number of directions (D) in calculation of sky-view factor; w,t,e: Hyperparameters in SAGA GIS module *Relative Heights and Slope Positions*.

GEOBIA offers the possibility to describe the objects themselves using shape metrics, and to quantitatively describe each object's relationships to its surroundings. For each object we derived shape metrics, mean values for land-surface variables, and textural features, as well as their mean values within adjacent objects (Table 1). We used two adjacency types: The Queen's contiguity from the *spdep* package [25], and the flow contiguity—an adjacency in flow direction of an object [55]. Overall, 43 features were used for landslide classification (ten land-surface variables, two textural features, seven shape metrics, twelve features in Queen's contiguity, twelve features in flow contiguity; Table A2).

### 2.3.4. Classification and Variable Importance

For landslide classification we used the support vector machine, a flexible supervised machine-learning technique [56]. SVM has become a widely used classification method that has shown its potential in landslide distribution modeling [7,13,21]. We used the *e1071* package in R [57] within the *mlr* modeling framework. The flexibility of SVM is controlled by hyperparameters that need to be tuned to achieve optimal classification results. We chose the radial basis function kernel and optimized the *gamma* and *cost* parameters in a random search with 250 iterations using a five-fold spatial cross-validation (within the training area) to also account for spatial autocorrelation effects [58]. Following Richter [59], *cost* was drawn between $2^{-12}$ and $2^{15}$, and *gamma* between $2^{-15}$ and $2^{6}$. As performance criterion we chose Cohen's Kappa (κ)—a well-established accuracy measure for classification problems [60], which was also applied in similar studies [13,15].

Since we split landslides into scarp and body, we performed a multiclass classification differentiating the three categories landslide scarp, landslide body, and non-landslide. The design of a suitable training and validation set for this step is challenging because the segmentation objects are not perfectly aligned with the inventoried landslides (see Section 3.2). To avoid a subjective manual assignment, we applied the following labelling procedure to automatically mark the objects in the training and validation sample as observed landslide or non-landslide. An object was defined to be a true landslide object if one of the following conditions was met:

$$\frac{\left| A_{obj} \cap A_{inv} \right|}{\left| A_{obj} \right|} \geq 0.6, \text{ or} \tag{1}$$

$$P_{obj} \in A_{inv}, \tag{2}$$

where $A_{obj}$ is the object in question, $A_{inv}$ is the intersecting inventoried landslide object, and $P_{obj}$ is the object's so-called 'point on surface' in the sense of [61]. All other objects obtained in the segmentation were labelled as non-landslides. Specifically, in the case of candidate scarp objects, the combined area of the inventoried landslide (body and scarp) was considered. For candidate body objects, only the body of the inventoried landslide was used. With this strategy, we ensured that minor secondary scarps or cracks in the landslide body that potentially have similar characteristics as the main scarp but were not included in the inventory, were not considered as part of the landslide body. To investigate the mapping error between the true landslide objects and the inventoried landslides, we calculated the positional mismatch (PM, 0% exact match, 100% no match) introduced in [62].

We trained and tuned the landslide detection model in the training area with a 5:1:1 ratio of non-landslide to landslide objects (body and scarp). The fitted model was then applied to predict class memberships on the validation set.

Machine-learning models are "black boxes". To gain insight into the contribution of the predictor variables to our model, we extracted the variable importance of each variable using a permutation-based assessment as proposed by Brenning et al. [63]. Using the *mlr* framework, we computed the mean decrease in κ from permuting the values of a variable and comparing them to predictions made on unpermuted data. Variables with a large decrease in κ are of greater importance in the model for detecting landslides.

### 2.3.5. Post-Processing of Predicted Landslide Objects

Since landslide bodies and scarps may be represented by multiple segmentation objects, we post-processed the SVM classification result with the aim to create complete, unique landslide objects. More specifically, this step ensured that landslide body parts were associated with a landslide scarp, and vice versa. However, we also retained isolated objects that had a high predicted probability of at least 0.8 for being any landslide part.

In the first post-processing step, all contiguous predicted scarp objects were merged while accounting for the information of the corresponding adjacency in flow direction. In the second step, these corresponding neighbors were examined and selected when they fulfilled the condition of being a predicted body. Subsequently, the second step was repeated until all neighbors in flow direction predicted as body were found. In the last step, all predicted adjacent landslide objects were merged to a single landslide object representing the post-processed classification result.

### 2.3.6. Accuracy Assessment

We assessed the accuracy of the landslide detection on the pixel- and the object-level. Additionally, as we applied a post-processing procedure on the landslide classification, we distinguished between the landslide classification with and without post-processing.

For the pixel-level assessment, we rasterized and overlaid the predicted landslide objects and the inventoried landslides, and assessed the accuracy using $\kappa$. In the object-level assessment, we intersected the predicted landslide objects with the inventoried landslides following Eeckhaut et al. [7]. According to [7], if at least 50% of the area of an inventoried landslide was covered by a predicted landslide object, it was counted as correctly identified.

## 3. Results

### 3.1. Mask and Segmentation

In the 'mask' step, the optimization for discriminating landslide body and scarp area using the F-score (beta = 3) yielded an optimal scale parameter $S = 16$ and a threshold $T = 5.5$ at an F-score of 0.153 (Figure 3). The F-score was relatively insensitive to scale factors between 14 and 30 and slope threshold values below 10° and above 4°.

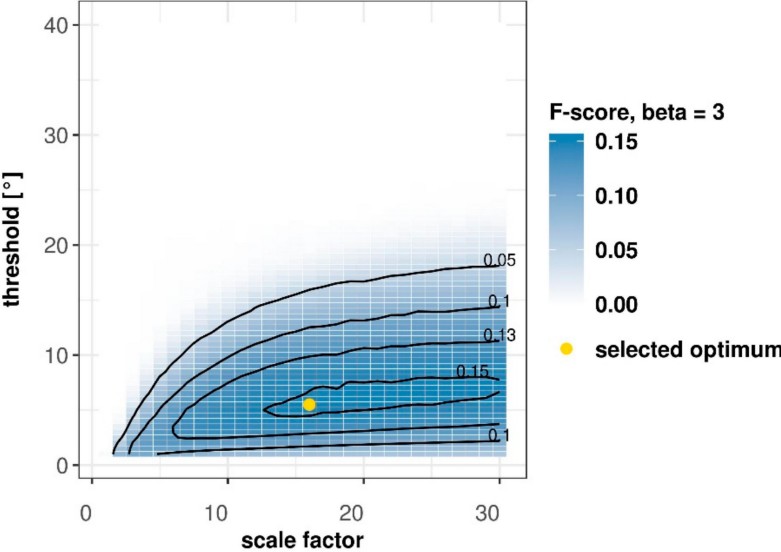

**Figure 3.** Optimization results for ´mask´ step: Performance of high-pass filtering and thresholding with different parameter values.

For the segmentation of the candidate scarp area in the 'segment' step, the OF showed multiple local optima in the *threshold* range of 0.065 to 0.095; threshold values < 0.02 were also nearly as good (Figure 4a). For segmenting the landslide body area, *Levels of Generalisation* between 2 and 3.75 were nearly optimal (Figure 4b). After visual inspection of the segmentation results with the locally optimal scales, we decided to use the *threshold* of 0.075 for the final segmentation of the candidate scarp area, and the *Level of Generalisation* of 3.25 for the final segmentation of the candidate body area. The reason for this selection was based on the desired object size: While for the candidate scarp area, a local optimum with a small threshold resulted in more suitably sized objects, for the candidate body area a relatively large value of *Level of Generalisation* resulted in adequately sized objects.

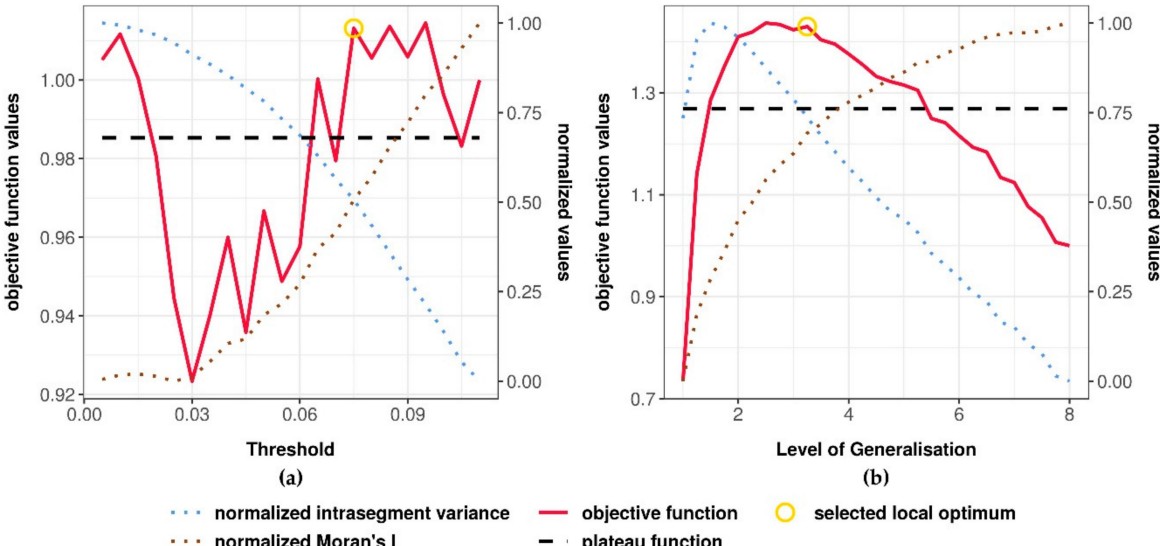

**Figure 4.** Optimization results for ´segment´ step: (**a**) Objective function of GRASS GIS segmentations of candidate scarp area optimizing the scale parameter *threshold*; (**b**) objective function of SAGA GIS segmentations of candidate body area optimizing the scale parameter *Level of Generalisation*; note that the y-axis scales of the objective function plots (a, b) are different to make the figures easier to read.

Applying these optimized parameters in the training and validation area (example in Figure 5), around 95% of the training area became a candidate body area (validation area: 94%); 5% was candidate scarp area (validation area: 6%). The segmentation generated 147,352 candidate landslide objects (validation area: 80,851), with 31,193 candidate scarp (validation area: 18,880), and 116,159 candidate body objects (validation area: 61,971). Using the proposed labelling procedure, 2477 of these candidate objects in the training area were labelled as landslide scarp (validation area: 1625), and 3728 as landslide body objects (validation: 2480), respectively. The PM was 27% in the training area and 22% in the validation area.

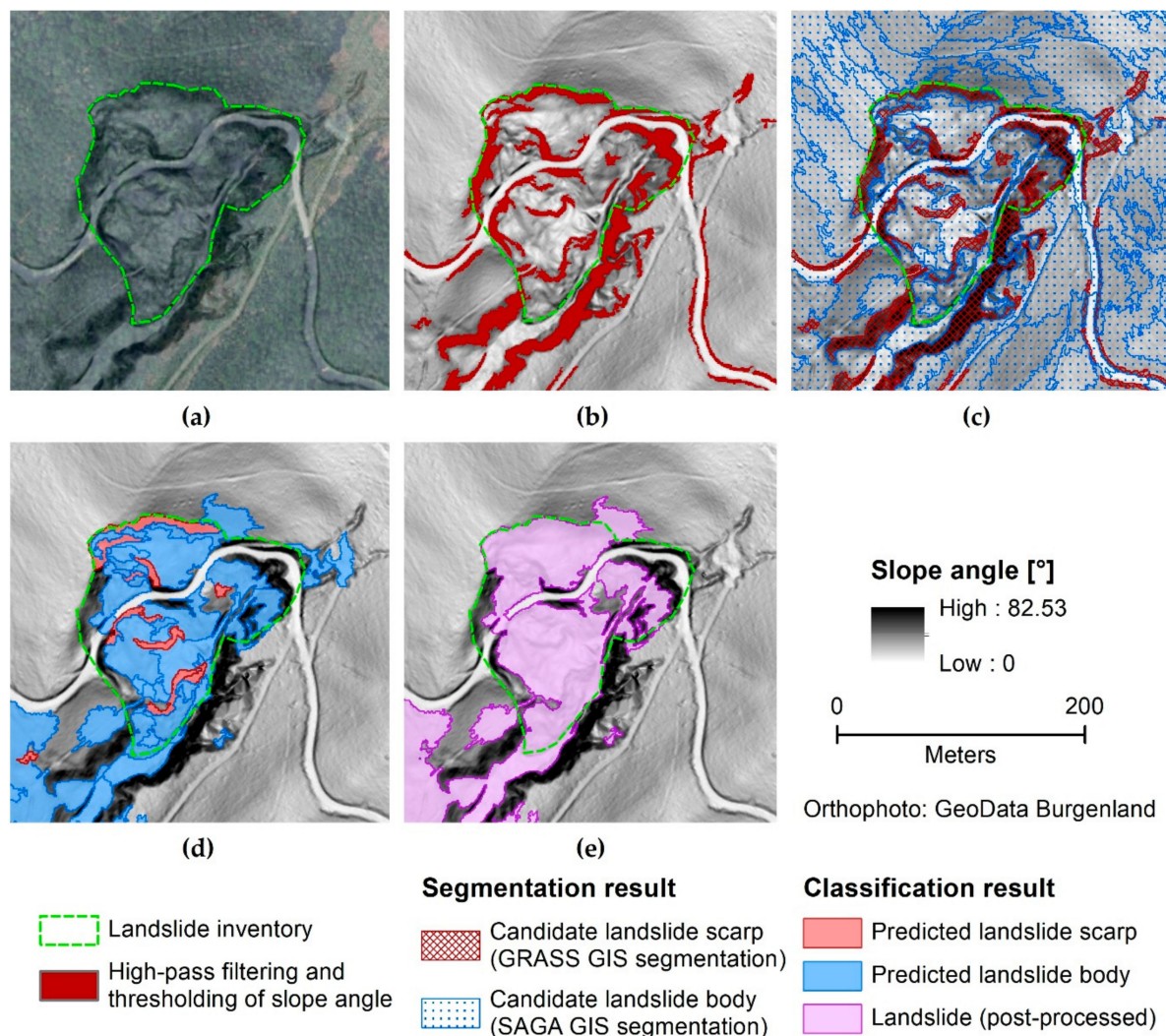

**Figure 5.** Results of the processing steps: (**a**) Landslide visible in a slope map; (**b**) 'Mask' step: Candidate landslide slope and body areas; (**c**) 'Segment' step: Segmentation result; (**d**) 'Classify' step: Predictions; (**e**) post-processed classification.

### 3.2. Landslide Classification, Accuracy, and Variable Importance

The tuning of the SVM classifier in the training area resulted in optimal hyperparameters of *cost* = 155 and *gamma* = 0.00117 at a mean κ of 0.55. The pixel-level accuracy of the classification in the validation area was lower with κ = 0.42, although post-processing increased it to 0.48. In the object-level validation, 69% of the inventoried landslides were successfully detected. This rate dropped slightly to 66% after post-processing.

Visual comparison of the classification predictions and the inventory maps showed a satisfactory agreement (Figure A1a). Large landslides with a strong geomorphometric signature were more often successfully detected than small landslides with a weak one. Generally, the automatically mapped landslide objects were rather wiggly than smooth in shape which resulted in a mismatch with the outline of inventoried landslides (Figure 5c–e). In addition, we found landslide objects extending outside the areas of the actual inventoried landslide (Figure 5c–e), which may be caused by limitations of the segmentation algorithms resulting in wiggly or coarse objects (see Section 4.3), misclassifications, or false assignments in the labelling procedure.

Our methodology did not explicitly account for anthropogenic features such as roads possibly intersecting a landslide, which may result in split or undetected landslides (Figure 5c–e). False positive landslide objects most commonly occurred in areas of fluvial erosion (Figure A1c), areas where forest tracks or gullies as relicts of past forestry activities were visible in the HRDTM (Figure A1b), and in built-up areas on hillslopes (Figure A1d). However, in this study we did not observe false-positive landslide objects along river banks, road ditches, or in alluvial deposits (Figure A1a). By post-processing the classification, we successfully reduced the number of false positives (increase of κ), but also eliminated true positives among the inventoried landslides (decrease of correctly classified inventoried landslides; Figure 5e).

The three most important variables in detecting landslides using the SVM were texture entropy in Queen's contiguity (κ decrease 0.40), slope angle (κ decrease 0.34), and texture entropy in flow contiguity (κ decrease 0.23). Generally, shape metrics had a lower importance; compactness as most important shape metric ranked 10th with a κ decrease of 0.11 (Figure 6, Table A2).

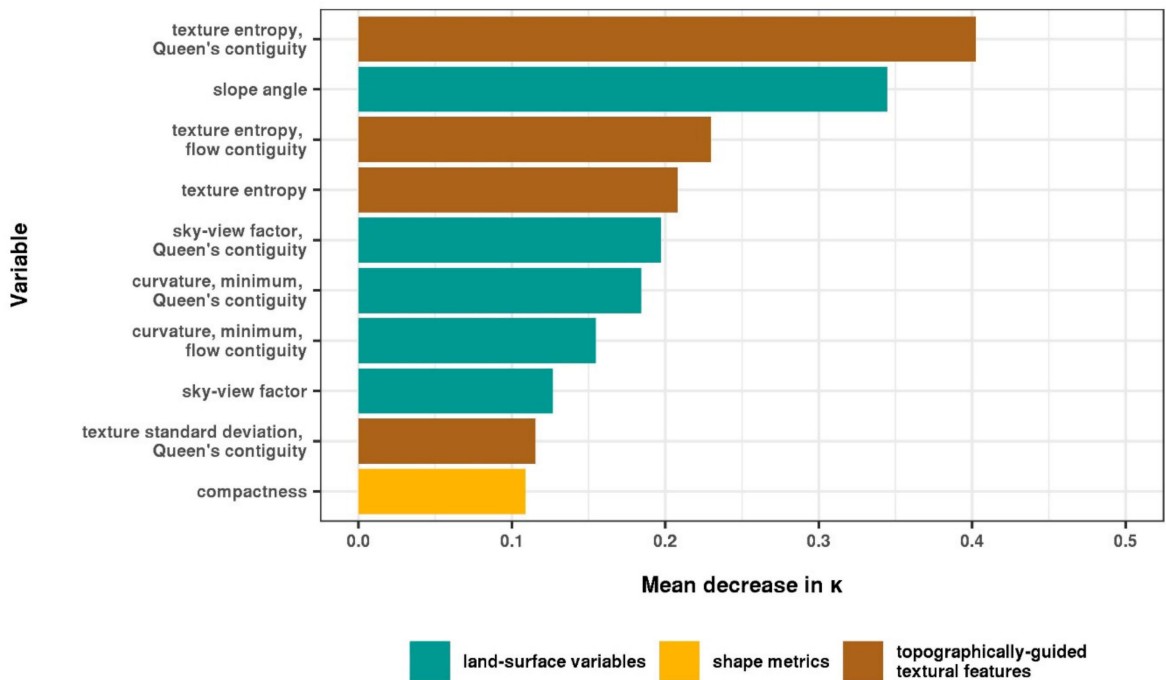

**Figure 6.** Result of variable importance: The ten most important variables assessed by permutation-based importance and measured in mean decrease in κ. Complete overview in Table A2.

## 4. Discussion

### 4.1. Classification Accuracy and Relevant Predictors

We successfully applied and assessed an automated open source landslide detection procedure solely based on HRDTM-derived data. The moderate classification agreement based on the pixel-level assessment (κ of 0.42, and 0.48 in post-processed classification) is consistent with previous research using similar methods [13,15] (κ between 0.45 and 0.6). Although different performance measures and data sets were used in studies within commercial software, they also showed a moderate predictive skill [7,18] (percentage of inventoried landslides detected correctly at the object level 69%, after post-processing 66%; 71% in [7] and 68.7% and 73.3% in [18]).

Assessing the accuracy of GEOBIA is a challenging task: First, there is no standard criterion to evaluate the quality of the segmentation [7]. Second, landslide inventories are never complete and can vary greatly depending on the persons digitizing them [5,7,62,64] (40–80% PM in [5] and 50–70% PM in [62]). Third, labelling an object as an observed landslide can be ambiguous when the object does not exactly match the mapped inventoried landslide. While the first and second mentioned challenges are still unresolved, we tried to overcome the third one by introducing an objective, automated labelling procedure for landslide and non-landslide objects. We achieved a low positional mismatch (training area: 27% PM, validation area: 22% PM) compared to the reported values in [5] and [62] confirming the success of the proposed labelling strategy.

While landslide scarps were often well captured by the ´filter-mask-segment´ approach, the landslide body was too complex to be easily delineated. The surface roughness, in this study represented through a roughness index [47], is often considered as a descriptive parameter for the landslide body [7,21,37]. However, this parameter showed only little importance in detecting landslides (κ decrease < 0.02, Table A2).

Eeckhaut et al. [7] showed a high discrepancy in accuracy between shallow and deep-seated landslide detection. In their study, shallow landslides were identified with a low detection rate (18.2%), suggesting that the geomorphometric signature is too weak. In our analysis we did not discriminate between shallow and deep-seated landslides, as we were lacking this information in the landslide inventory. Extracting the information on the depth of the sliding surface based solely on HRDTM is a challenging task [65,66]. Yet, we suggest that model performance may benefit from further discrimination between deep-seated and shallow landslides or between different landslide types producing contrasting geometries (e.g., translational landslide versus debris flow).

Regarding the variable importance, especially the textural features of entropy showed high importance for landslide detection (κ decrease > 0.21, see Table A2), confirming previous findings by Stumpf and Kerle [19]. However, this appears to contrast with other studies that reported no significant gain in accuracy [15,21]. This apparent contradiction may be due to subtle differences in the chosen algorithms. Specifically, topographically guided texture features calculated in flow direction (as in [19] and this study) have shown more promising results than textural features calculated for all pixels of an object [15,21]. We therefore encourage the use of textural features calculated in flow direction for landslide detection in future studies. Apart from this, the low importance of shape metrics (rank ≤ 10) was surprising, since the expected importance of object-describing features [7] such as a semi-circular scarp (κ decrease 0.02) or a scarp object orientation perpendicular to its flow direction (κ decrease 0.02) could not be confirmed. Less surprising was the high importance of slope angle (κ decrease 0.35).

Other studies included more statistical features and object statistics (e.g., standard deviation, maximum and minimum values, etc.) for landslide detection, but without precisely mentioning its benefit [7,15]. In the proposed procedure we extracted only object mean values for underlying land-surface variables and textural features. In the next step, we could also enhance the feature space and assess their importance in detecting landslides. Combined with the use of feature selection methods, an optimized data set could then be created that may further improve the landslide detection rate.

In addition, there are other adjusting screws which can be refined and some of which have already been investigated: Pawłuszek et al. [15] examined different HRDTM resolutions as basis for their GEOBIA approach and found the highest accuracies between 2 and 5 m resolution. In addition, some features used in this study are scale-dependent (e.g., openness, see Setting in Table 1). This scale dependency may be expressed by different moving window sizes, search radii, or other scale-controlling parameters, while computing the input features. A quantitative analysis for finding optimal scale values of the input features may enhance the model fit [13,66]. However, these optimizations come along with additional computational cost which could not be accomplished in this analysis.

## 4.2. Optimization and Transferability

In GEOBIA, the segmentation and classification techniques require input parameters (scale and hyperparameters) controlling the result. In numerous studies, the segmentation scale parameters are subjectively estimated by a trial-and-error method [21], classifier default settings are used [7,15], or parameters are estimated based on non-spatially partitioned training data [7,15,21], which may lead to biased, mostly over-optimistic results [58]. In our analysis we avoided bias-reduced results by optimizing our procedure only in the training area and by evaluating the classification in the validation area. In the training area, we selected segmentation scale parameters using an objective function, and performed spatial hyperparameter tuning of the classifier to (1) avoid over-optimistic classification results due to spatial autocorrelation effects, (2) find optimal model or function settings beyond default or arbitrarily chosen values, and (3) to avoid subjective and time-consuming trial-and-error methods. We strongly recommend that similar tuning strategies are adopted in future studies.

The transferability of the proposed landslide detection procedure needs to be tested in other regions. We suppose that landslides leaving a clear geomorphological signature and identifiable characteristics such as a landslide scarp and body in the HRDTM, can be detected in other regions. Limitations exists for landslide types without these characteristics (e.g., falls, topples, creeps, cf. [20]).

## 4.3. Potential and Limitations of Open Source Software

With the help of open source GIS software such as GRASS or SAGA GIS, we were able to derive the land-surface variables most commonly used in landslide detection studies. This software also offers algorithms for image segmentation. By leveraging the growing number of programming interfaces from within the free development environment R, it was possible to automate, combine, and extend geocomputing tools that are not currently available in a single software. We were thus able to exploit the state-of-the-art machine-learning capabilities offered by the *e1071* and *mlr* packages in R as part of the proposed workflow.

This study also revealed limitations of available open source GIS software that was used for segmentation, a critical step in our analysis. Therefore, the segmentation algorithms missed important adjusting screws to fine-tune the resulting objects: For example, SAGA's module did not provide control over the minimum object size; this required a workaround by sieving and expansion. Moreover, the "wigglyness" in the shape of the resulting objects could not be controlled by any settings. The algorithm available in SAGA GIS created artefacts in no-data areas but was around 10 times faster than its GRASS GIS counterpart, which is why we had to combine two different segmentation approaches implemented in different GIS software. In addition, the segmentation of multiple input images was limited in the sense that no weighting of images was possible, and further information as thematic layers (such as rivers or streets; for further splitting of image objects) could not be added. Furthermore, some open source functions are still missing a clear, comprehensive user guide (e.g., SAGA GIS module *Seeded Region Growing*).

## 5. Conclusions

Inventories are the basis for any landslide analysis, but their digitization by human experts is a time-consuming process [3] and is subjective to the opinion of the creator [5]. In recent years, multiple semi- or fully automated landslide detection approaches emerged to support landslide mapping. However, these approaches are often based on commercial software and lack reproducibility due to restricted software access or unreleased source code. In addition, these approaches commonly rely on optical data leading to dependence on weather situation and vegetation cover [16–19].

Our proposed approach was completely accomplished with open source software and solely based on airborne LiDAR HRDTM-derived data. We successfully applied different optimization procedures (e.g., on segmentation scale parameters and classifier hyperparameters) to avoid biased results and trial-and-error methods. Our classification achieved a moderate κ of 0.42 (post-processed: 0.48) and 69% of correctly detected inventoried landslides (post-processed: 66%) similar to performances of approaches using commercial software products (κ between 0.45 and 0.6 in [13,15]; 68% to 73% of correctly detected inventoried landslides in [7,18]). We found that topographically guided texture features showed high importance for landslide detection (textural features of entropy with κ decrease > 0.21), and were surprised about the low importance of shape metrics (rank ≤ 10).

The transferability of the developed approach needs to be tested in other regions, but we suppose that worldwide landslides with a clear geomorphological signature in the HRDTM will be identifiable with this approach. Nevertheless, the automated compilation of landslide inventories still requires a thorough assessment by experts and cannot replace validation based on field observations.

**Supplementary Materials:** The following are available online at http://www.mdpi.com/2220-9964/8/12/551/s1. Sample data and code: R-Project Example.zip.

**Author Contributions:** Conceptualization, Raphael Knevels, Helene Petschko, Philip Leopold, and Alexander Brenning; Formal analysis, Raphael Knevels; Methodology, Raphael Knevels, Helene Petschko, and Alexander Brenning; Resources, Philip Leopold and Alexander Brenning; Software, Raphael Knevels; Supervision, Helene Petschko and Alexander Brenning; Validation, Raphael Knevels; Writing—original draft, Raphael Knevels; Writing—review and editing, Raphael Knevels, Helene Petschko, Philip Leopold, and Alexander Brenning.

**Funding:** This research received no external funding.

**Acknowledgments:** We are grateful to the Federal State of Burgenland for providing the high-resolution digital elevation model. Furthermore, we would like to thank four anonymous reviewers for their valuable comments.

**Conflicts of Interest:** The authors declare no conflict of interest.

## Appendix A. Descriptive Summary of Input Data

**Table A1.** Summary of the landslide inventory data.

| Municipality | | Training Area | | | Validation Area | | |
|---|---|---|---|---|---|---|---|
| | | Piringsdorf, Steinberg-Dörfl | | | Unterrabnitz-Schwendgraben, Pilgersdorf | | |
| Number | landslides | | 184 | | | 198 | |
| | scarps | | 248 | | | 302 | |
| | bodies | | 184 | | | 199 | |
| | | *min* | *mean* | *max* | *min* | *mean* | *max* |
| Area [m$^2$] | landslides | 121 | 4642 | 148,122 | 270 | 7899 | 74,860 |
| | scarps | 11 | 328 | 2918 | 14 | 560 | 4095 |
| | bodies | 95 | 4175 | 141,972 | 6 | 7010 | 68,232 |
| Perimeter [m] | landslides | 44 | 239 | 1907 | 62 | 361 | 1847 |
| | scarps | 15 | 141 | 788 | 18 | 181 | 736 |
| | bodies | 43 | 231 | 2090 | 54 | 353 | 1850 |

## Appendix B. Further Presentation of Results

**Table A2.** Overview of variable importance.

| Variable | | Decrease in κ | Rank |
|---|---|---|---|
| *land-surface variables* | | | |
| curvature | maximum | 0.019 | 30 |
| | flow contiguity | 0.009 | 40 |
| | Queen's contiguity | 0.010 | 39 |
| | minimum | 0.007 | 42 |
| | flow contiguity | 0.155 | 7 |
| | Queen's contiguity | 0.184 | 6 |
| | plan | 0.011 | 38 |
| | flow contiguity | 0.014 | 34 |
| | Queen's contiguity | 0.025 | 20 |
| | profile | 0.020 | 25 |
| | flow contiguity | 0.019 | 29 |
| | Queen's contiguity | 0.035 | 16 |
| flow accumulation | | 0.011 | 36 |
| | flow contiguity | 0.013 | 35 |
| | Queen's contiguity | 0.015 | 33 |
| normalized height | | 0.003 | 43 |
| | flow contiguity | 0.011 | 37 |
| | Queen's contiguity | 0.017 | 31 |
| openness | | 0.032 | 19 |
| | flow contiguity | 0.021 | 22 |
| | Queen's contiguity | 0.037 | 15 |
| roughness index | | 0.020 | 27 |
| | flow contiguity | 0.017 | 32 |
| | Queen's contiguity | 0.008 | 41 |
| sky-view factor | | 0.127 | 8 |
| | flow contiguity | 0.033 | 17 |
| | Queen's contiguity | 0.197 | 5 |
| slope angle | | 0.345 | 2 |
| | flow contiguity | 0.021 | 23 |
| | Queen's contiguity | 0.107 | 11 |
| *textural features in flow direction* | | | |
| entropy | | 0.208 | 4 |
| | flow contiguity | 0.230 | 3 |
| | Queen's contiguity | 0.402 | 1 |
| standard deviation | | 0.056 | 14 |
| | flow contiguity | 0.115 | 9 |
| | Queen's contiguity | 0.064 | 12 |
| *shape metrics* | | | |
| Compactness | | 0.109 | 10 |
| Convexity | | 0.021 | 24 |
| interior edge ratio | | 0.023 | 21 |
| length-to-width ratio | | 0.020 | 26 |
| maximum distance between a polygon's vertices | | 0.033 | 18 |
| object orientation to flow direction | | 0.020 | 28 |
| shape index | | 0.061 | 13 |

κ: Cohen's Kappa.

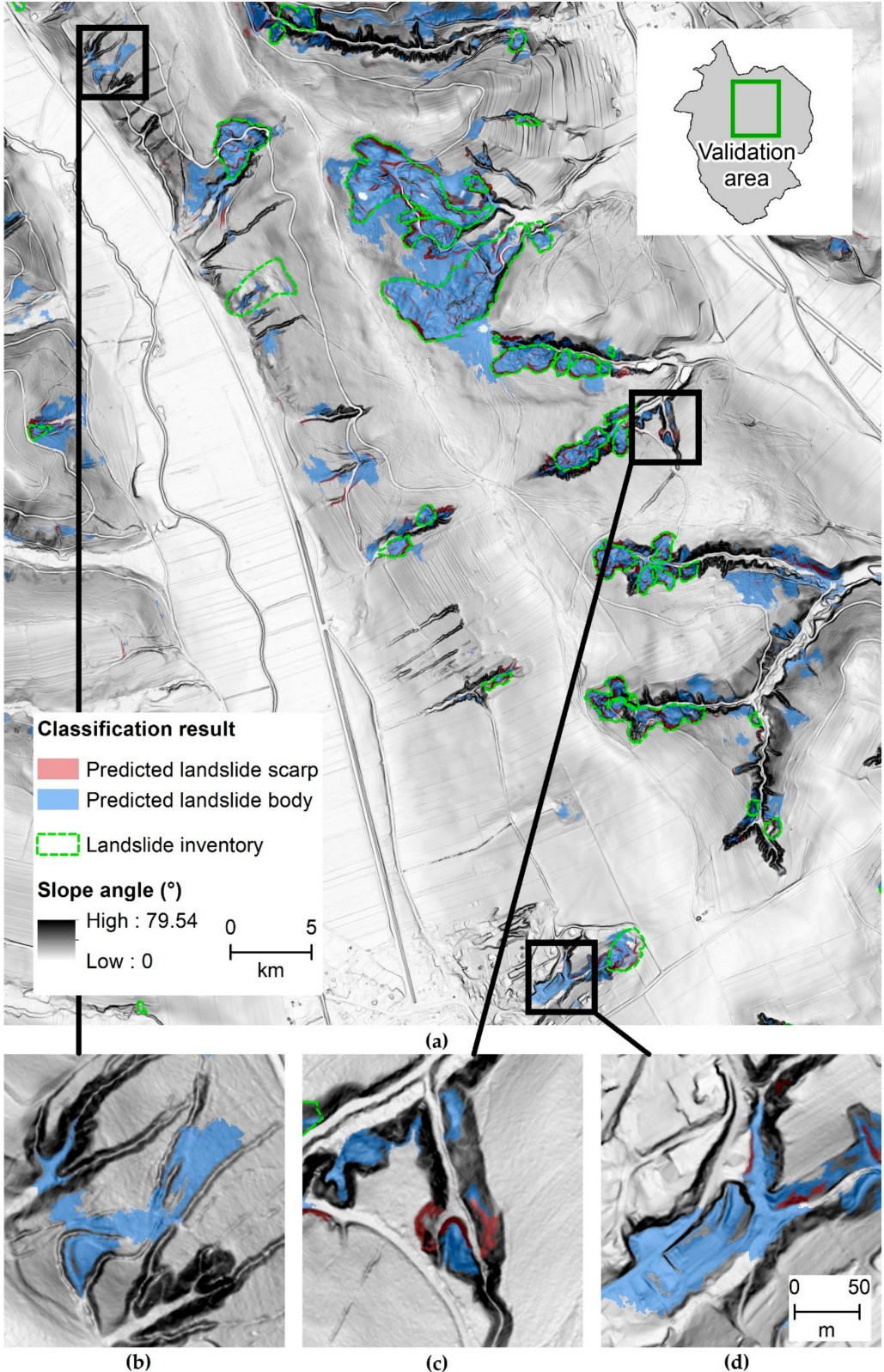

**Figure A1.** Examples of predictions in validation area: (**a**) Overview with false positive examples; (**b**) misclassification due to forestry tracks and gullies; (**c**) misclassification in fluvially eroded areas; (**d**) misclassification in built-up areas.

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
