# Peer review of "Geographic Object-Based Image Analysis for Automated Landslide Detection Using Open Source GIS Software"

_ijgi, doi:10.3390/ijgi8120551_

Round 1
Reviewer 1 Report
The paper contributes to the extensive literature on application of machine learning in mapping natural hazards features. The authors highlight the novelty of their approach in terms of open source software implementation and several original ideas in their methodology. The accuracy of the approach is comparable with other methods in the literature so the major improvement appears to be the open source implementation. The study also contributes some insights on importance of various elevation surface metrics in relation to landslides. The paper is well written so I have just few minor comments:
1. The accuracy assessment was performed using manually delineated landslides from the DEM without field validation - this assumes that no field data were available. Although this is a valid approach is there any possibility that some of the areas identified as false positives were actually landslides missed by visual mapping? This issue is included in the discussion as unresolved but at least based on literature, what would be the contribution of the visual mapping error to the mismatch ?
2. To avoid confusion with terrestrial lidar it would be useful to add the word airborne lidar on line 113.
3. Paper with comparable results and probably simpler approach for slightly different type of landslides - debris flows: Lyons, N.J., Mitasova, H., Wegmann K.W., 2014, Improving mass-wasting inventories of debris flow-dominated channels in non-glaciated terrains, Landslides 11(3), pp. 385-397. DOI 10.1007/s10346-013-0398-0.
4. I was wondering why did the authors had to use Taudem, given that there is a wide range of flow accumulation algorithms available in GRASS and its addons which would make it possible to run it from R.
Reviewer 2 Report
The manuscript topic fits very well with the scope of the journal. The authors investigated the potential of Open-Source software for automatic landslide detection using HRDTM. The manuscript is well written, the introduction provides a sufficient background, the methodology is clear and robust, and the results are innovative. The work is scientifically sound and it deserves to be published in IJGI. However, the conclusion is very short and incomplete; therefore, I strongly suggest the authors to rewrite this part of the manuscript before publication.
Reviewer 3 Report
This paper used a classification approach, geographic object-based image analysis (GEOBIA), to predict landslide scarp and body with high-resolution digital terrain models. R and related open-source geographic information system software were used in this study. This is helpful for decision makers to identify potential landslide areas at a low cost as data are accessible, and the software are free to use. My comments are listed below.
First, I am confused by the second step, segmentation, where two methods are used. What is the relationship between the two segmentation methods in the two platforms (SAGA GIS and GRASS GIS)? See minor comments about Line 221
Second, the section of accuracy evaluation for object-level assessment should be improved. See minor comments about Line 307. This part is not clear. Besides kappa, there are other metrics/indices (e.g., producer accuracy, user accuracy…) available to evaluate the pixel-level classification accuracy.
Third, it is concluded that this approach “achieves accuracies comparable to performances of commercial software products” (line 468-469). However, I cannot find the comparison experiment with other commercial software. Further, the conclusion section should be further expanded.
Minor comments
Line 12: increased — increasing?
Line 66: what is post-failure characteristics? Citations?
Line 70: what are the limitations calling back to? Post-failure characteristics and failure to detect landslides in forest? Does HRDTM solve them?
Figure 1: north arrow could be somehow smaller
Line 104: area—areas
Line 105: increased — increasing
Line 181: high-pass filter is to filter out high-frequency signals. How is slope filtered out with this?
Line 217: combine the results from two algorithms or combine the two algorithms?
Line 221: what is larger? Confused. The authors claimed SAGA GIS is faster and GRASS GIS can ignore no-data areas, but SAGA cannot. Does this mean those large areas to be paralleled in SAGA GIS do not have masked area (no-data) so that they do not have to be done in GRASS GIS?
Line 228: why 50 square meters to avoid over-segmentation? Did the authors assume the landslide body area should be larger than the identified area? Is that from literature or empirical knowledge?
Line 303: what does it mean by distinguishing “between the landslide classification with and without post-processing”?
Line 307: based on what that the authors made such an assumption that 50% coverage of predicted landslide on the inventoried landslide is correct. Is there any evidence or ideas to support the validity of this threshold? It is important because that is related to the quality control.
Line 335: add thousand separator to the number 2,477
Line 336: add thousand separator to the number: 1,625, 3,728
Line 468: The comparison with commercial software was not found
Reviewer 4 Report
The manuscript presents a study on the automation of landslide vulnerability analysis using high-resolution digital terrain models and open-source software. Though the study used standard tools and methods, the methods were integrated in an interesting manner. The authors also make a good presentation of their study by comparing it with the existing studies in the literature. They also mentioned the possible limitations in applying the method in other locations. The authors should consider the following concerns in improving the manuscript.
In line 245, it is mentioned that the process was carried out at the pixel level unlike what is reported in the literature. Why did they adopt this approach and what is the effect on the study? No explanation has been given for the extension of the classified landslide beyond the areas of the inventoried landslide (Figure 5d and 5e). Though the standard of English is okay, the authors need to do minor editing. For example, in line 352 "In addition, we found landslide objects partly containing areas outside of the actual inventoried landslide..." can be better expressed as "In addition, we found landslide objects extending outside the areas of the actual inventoried landslide..".Author Response
Please see the attachment

Round 2
Reviewer 3 Report
My comments were addressed.